# Maturation Stress and Wood Properties of Poplar (*Populus × euramericana* cv. 'Zhonglin46') Tension Wood

**Yamei Liu** [1,2] , **Xiao Wu** [1] , **Jingliang Zhang** [1] , **Shengquan Liu** [1] , **Katherine Semple** [2] **and Chunping Dai** [2,*]

1   Key Laboratory of State Forest and Grassland Administration on "Wood Quality Improvement & High Efficient Utilization", School of Forestry & Landscape Architecture, Anhui Agricultural University, Hefei 230036, China; liuyamei3980@126.com (Y.L.); banban199609@gmail.com (X.W.); 1169076986@stu.ahau.edu.cn (J.Z.); liusq@ahau.edu.cn (S.L.)

2   Department of Wood Science, Faculty of Forestry, University of British Columbia, Vancouver, BC V6T1Z4, Canada; katherine.semple@ubc.ca

*   Correspondence: chunping.dai@ubc.ca

**Abstract:** Understanding the maturation stress and wood properties of poplar tension wood is critical for improving lumber yields and utilization ratio. In this study, the released longitudinal maturation strains (RLMS), anatomical features, physical and mechanical properties, and nano-mechanical properties of the cell wall were analyzed at different peripheral positions and heights in nine artificially inclined, 12-year-old poplar (*Populus × euramericana* cv. 'Zhonglin46') trees. The correlations between the RLMS and the wood properties were determined. The results showed that there were mixed effects of inclination on wood quality and properties. The upper sides of inclined stems had higher RLMS, proportion of G-layer, bending modulus of elasticity, and indentation modulus of the cell wall but a lower microfibril angle than the lower sides. At heights between 0.7 m and 2.2 m, only the double-wall thickness increased with height; the RLMS and other wood properties such as fiber length and basic density fluctuated or changed little with height. The RLMS were good indicators of wood properties in the tension wood area and at heights between 0.7 m and 1.5 m. The results of this study present opportunities to better understand the interactions and effects of these two phenomena, which both occur quite frequently in poplar stands and can influence the wood quality of valuable assortments.

**Keywords:** released longitudinal maturation stresses; wood properties; *Populus × euramericana* cv. 'Zhonglin46'; tension wood; peripheral positions; heights

## 1. Introduction

Tree growth stress caused by inner older xylem tissues restricting shrinkage of outer new cells leads to complex distribution of mechanical stresses during tree growth [1,2]. It presents in all trees, with variable magnitude, helping enhance the strength of tree trunks and branches and controlling their growth orientation against disturbance by gravity and wind [3]. Growth stresses also act as a pre-stressing system which can reduce the risk of axial buckling of fiber [4].

The thickening in the secondary cell wall generates mechanical stresses at the periphery of the trunk mostly along the stem axis [5]. When reaction wood is produced in inclined stems and/or branches, the distribution of stress across the cross section is heterogeneous [6–8], whereby greater stress on one side generates a bending moment to facilitate gravitropism in stems [9]. Eccentric growth and stiffness heterogeneity help to optimize this process [10]. In angiosperms such as *Populus*, tension wood is a reaction to environmental conditions generated on the upper side to restore the trunk to the upright position [11,12]. Tension wood is characterized by more xylem fiber cells, thicker secondary walls, higher cellulose content, and less lignin biosynthesis [13]. Some temperate-zone

hardwood species such as *P. euramericana* contain a gelatinous (G) layer with the characteristics of high tensile growth stresses and low microfibril angle (MFA) values [14] which in turn affects wood strength properties and quality.

Growth stress affects timber production, causing problems such as radial splitting at the log edges and bowed/twisted sawn planks [15,16]. Some reaction wood can cause serious defects, leading to economic losses in the lumber industry [3]. Understanding tree growth stresses and reaction wood, their distribution, and associated wood properties is important not only for ecophysiology but also in the fields of wood quality, timber conversion, and timber engineering.

Numerous studies have highlighted the influence of growth-stress levels on the properties of tension wood in poplar [17], beech [18], eucalyptus [19], and chestnut [20], with most focusing on the microscopic and macroscopic levels. Very few studies report on the relationship between growth stress and wood properties at the nanometer level of tension wood. Nano-indentation can be used to characterize mechanical properties of wood such as indentation modulus and stiffness of the cell wall [21–24].

Poplars (*Populus* sp.) are short-rotation and highly tolerant hardwoods that can grow on poor sites and soil conditions and are widely distributed across the Northern Hemisphere [25–27]. Poplars have become commercially important in China, with several clones such as *P. euramericana* cv. 'Zhonglin46' being commonly used in the plywood, solid wood, and paper industries [28,29]. However, their stems are susceptible to tension wood formation [17,30]. It is therefore necessary to assess the levels and distribution of growth stresses to increase sawn-lumber yields and utilization ratio. Growth stress can be easily measured experimentally using the release method [31]. In this study the released longitudinal maturation strains (RLMS), anatomical features, physical and mechanical properties, and nano-mechanical properties of the cell wall were examined in leaning trees of poplar (*P. euramericana* cv. 'Zhonglin46'). The objectives of this research were to (1) identify the changes in maturation stresses and related wood properties along peripheral and vertical positions and (2) determine and discuss the correlations between RLMS and wood properties.

## 2. Materials and Methods

### 2.1. Study Site and Tree Information

The experimental site was located at Anhui Agricultural University in Hefei City, Anhui province, China (31°87′ N, 117°26′ E), with a subtropical humid climate and monsoon influence. The site is 20–40 m ASL, with annual temperature range of 13–22 °C, average annual temperature of 15.7 °C, annual rainfall range of 794–1523 mm, and average annual rainfall of 1000 mm. The sampled poplar trees were spaced in a 3 m × 3 m configuration in a monoculture planted on a north-facing slope with soil type being yellow-brown soil.

In January 2021, a total of nine well-developed 12-year-old trees were selected for sampling. Figure 1 shows typical trees sampled. The trees were planted in 2009 and were artificially inclined by being fixed to 15–60° inclined iron braces for a year to test the growth stress of seedings for another experiment. After the first year, the braces were removed, and the trees were allowed to continue growing untethered. The average tree height, DBH, and tilted angle of the sampled trees are listed in Table 1.

### 2.2. Released Longitudinal Maturation Strains

RLMS were measured using strain gauges [31,32] on seven measuring points of the leaning tree at three different heights, viz., 0.7 m, 1.5 m, and 2.2 m from the ground (Figure 1a). In each sampled tree, the uppermost side was set as position 0°, and seven positions were numbered in clockwise sequence at 50–60° intervals, namely at 0°, 50°, 100°, 150°, −160°, −110°, and −60° around the trunk (Figure 2).

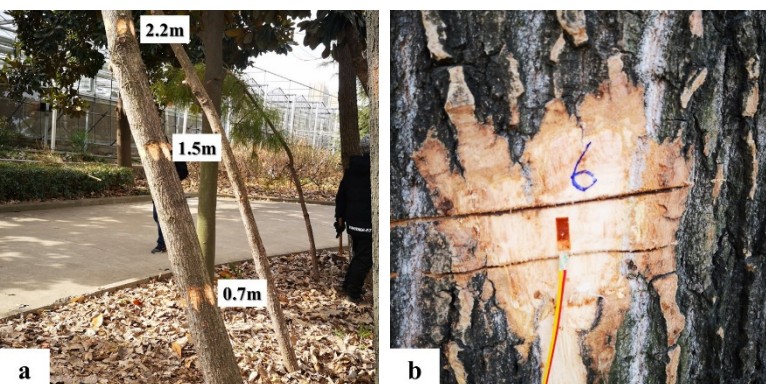

**Figure 1.** Image of sampled trees. (**a**) Measuring points at three different heights of tree 5; (**b**) maturation strains released by strain gauges of tree 6.

**Table 1.** Sample information for nine studied trees.

| Tree Number | Tree Height (m) | DBH (cm) | Tilt Angle from Vertical Position (°) |
|---|---|---|---|
| 1 | 19.6 | 38.8 | 27 |
| 2 | 21.7 | 51.2 | 11 |
| 3 | 18.5 | 35.6 | 17 |
| 4 | 16.5 | 23.8 | 15 |
| 5 | 15.9 | 18.4 | 24 |
| 6 | 14.3 | 15.2 | 18 |
| 7 | 16.2 | 21.0 | 13 |
| 8 | 14.1 | 14.7 | 16 |
| 9 | 15.8 | 19.1 | 20 |

DBH: diameter at breast height.

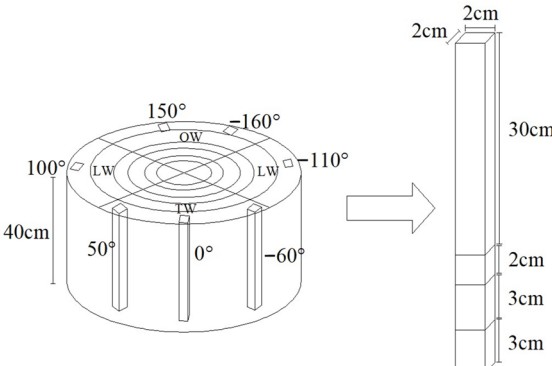

**Figure 2.** Schematic of measurement positions. TW: tension wood area (0°, 50°, and −60°); LW: lateral wood area (100° and −110°); OW: opposite wood area (150° and −160°).

Before testing, bark and cambium were removed using a knife and axe to expose a fresh and smooth xylem zone with a size of about 5 cm × 5 cm. A 10 mm long electric-wire strain gauge (Sigmar Co., BSF120-6AA-T, Jinan, China) was glued securely to the exposed zone and was connected to a strain meter (Sigmar Co., ASMB2-16, Jinan, China). Grooves 5 mm in depth were then cut using a handsaw a distance of 5 mm above and below each strain gauge (Figure 1b), and RLMS values were recorded until the changed strain values remained stable.

### 2.3. Sampling of Felled Trees

After RLMS measurement, all sampled trees were felled for wood properties sampling. Eccentricity in diameter was calculated as:

$$\text{Eccentricity} = \frac{T_{TW} - T_{OW}}{T_{TW} + T_{OW}} \tag{1}$$

where $T_{TW}$ was the radius of the TW area and $T_{OW}$ was the radius of the OW area.

Seven locations beneath the RLMS measurement positions at three different heights were marked on the trunk surfaces for further studies, and 21 wood properties samples were cut from each tree. Before sawing, the surfaces of disks were brushed with zinc chloro-iodide solution to detect tension wood areas [33]. Rectangular wood strips measuring 380 mm (longitudinal direction, L) × 20 mm (radial direction, R) × 20 mm (tangential direction, T) were cut along the axial of the stem. Specimens were measured for standard physical and mechanical properties according to Chinese National Standard GB/T 1929-2009 [34]. Bending modulus of rupture (MOR) and bending modulus of elasticity (MOE) were measured on specimens measuring 300 mm (L) × 20 mm (R) × 20 mm (T) and basic density (BD) specimens were 20 mm (L) × 20 mm (R) × 20 mm (T). Two 30 mm (L) × 20 mm (R) × 20 mm (T) samples were cut to test the compressive strength (CS) and anatomical features of the wood at each location.

### 2.4. Anatomical Features

Fiber length (FL) was determined from 30 measurements of macerated fibers at each location.

The specimens used to assess wood anatomical features measured 20 mm (L) × 10 mm (R) × 10 mm (T) and were pre-softened by microwaving at medium to high power for 5–10 min [27]. Transverse sections 20 μm in thickness were sliced using a rotary microtome (Leica, RM 2265, Wetzlar, Germany) and double-stained with safranine and Astra-blue [35] capable of detecting G fibers in tension wood (Figure 3a). Sections were observed, and thirty measurements each for double-wall thickness excluding the G-layer (2WT) were made on each transverse section. The proportion of G-layer (PG) was calculated on 20 randomly selected areas of 0.16 mm$^2$.

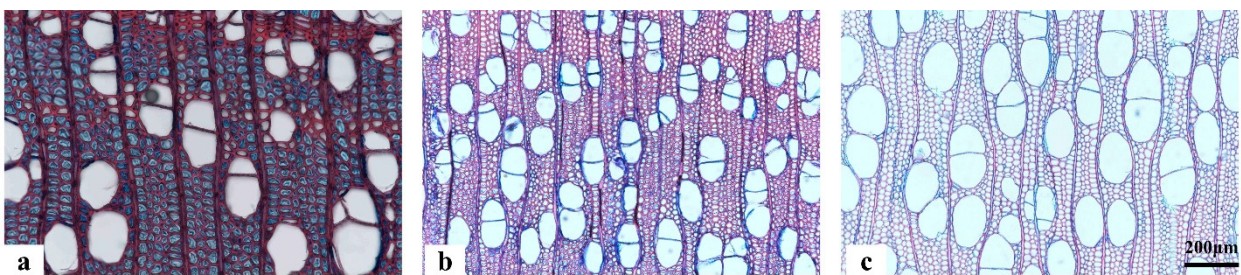

**Figure 3.** Light microscope view of anatomical sections of poplar tension wood. (**a**) TW area; (**b**) LW area; (**c**) OW area.

MFAs were measured by X-ray diffraction (Persee, XD-3, Beijing, China), on specimens measuring 30 mm (L) × 0.5 mm (R) × 10 mm (T). The recorded data were then processed using the 0.6 T method [36].

### 2.5. Physical and Mechanical Properties

BD of wood specimens (20 × 20 × 20 mm$^3$) was measured according to Chinese National Standard GB/T 1933-2009 [37].

MOR and MOE were tested on specimens measuring 300 × 20 × 20 mm$^3$ according to Chinese National Standard GB/T 1936.1, 1936.2-2009 [38,39]. Specimens measuring 30 × 20 × 20 mm$^3$ (LRT) were tested for CS according to Chinese National Standard GB/T

1935-2009 [40]. The mechanical properties were tested using a universal mechanical testing machine (Instron, 68TM-5, Boston, MA, USA).

### 2.6. Nano-Indentation Test

The indentation modulus of the cell wall (IM) was tested at different areas and heights. Three peripheral-position specimens were chosen at a height of 1.5 m: TW specimen from position 0°, LW specimen from position 100°, and OW specimen from position −160°. Position 0° was sampled at three heights: 0.7 m, 1.5 m, and 2.2 m.

Small wood blocks measuring 3 mm × 1 mm × 1 mm (LRT) were cut and equilibrated to 12% MC [24]. The blocks were then embedded in Spurr resin (SPI-Chem, West Chester, PA, USA). The blocks were then cut into a pyramid shape and the top portion was polished with a semi-thin microtome (Leica, RM 2265, Wetzlar, Germany) using a glass knife and a diamond knife (SYM6050H, Tokyo, Japan). The nano-indenter (Bruker, Dimension ICON, Billerica, MA, USA) and the Peak Force Quantitative Nano-mechanical Mapping (PF-QNM) mode were used. A load-shift curve was obtained using the Derjaguin–Muller–Toporov (DMT) contact model [41]. The indentation modulus distributions of the S1, S2, and G-layers were obtained using Nanoscope Analysis software v1.8.

### 2.7. Statistical Analyses

The statistical software SPSS 19.0 (IBM, New York, NY, USA) was used to conduct ANOVA to determine significant differences in wood properties among peripheral positions and tree heights, followed by Duncan's means comparison tests at the 5% significance level. Correlation coefficients between RLMS and wood properties were calculated by correlation analysis.

## 3. Results

### 3.1. Released Longitudinal Maturation Strains

The RLMS values in all nine trees were negative, meaning that longitudinal maturation stress was always tensile (Table 2). As can be seen in Table 2, average RLMS values for positions and heights ranged from −0.16% to 0.00% with an overall mean of −0.04% (±0.03 SD). Variation between peripheral positions and heights was large, with an overall CV of 75%. ANOVA analysis detected significant differences in the RLMS among different peripheral positions ($p < 0.001$), but no significant differences among different heights ($p > 0.05$). Distribution of the RLMS for peripheral positions and heights is shown in Figure 4. The upper sides (0° locations) had the largest RLMS, and the lower sides (corresponding to 150° or −160°) had the smallest values. Mean RLMS values changed little with height.

**Table 2.** Basic statistics for measured parameters.

| Parameters | *n* | Average | SD | Max | Min | CV% |
|---|---|---|---|---|---|---|
| RLMS (%) | 189 | −0.04 | 0.03 | −0.16 | 0.00 | 75.0 |
| FL (μm) | 7990 | 1302 | 227.71 | 2343 | 508 | 17.5 |
| 2WT (μm) | 8698 | 5.73 | 1.34 | 11.69 | 2.50 | 23.4 |
| MFA (°) | 294 | 14.90 | 6.83 | 34.10 | 0.10 | 45.8 |
| PG (%) | 189 | 20.3 | 22.5 | 68.8 | 0.0 | 110.9 |
| BD (g/cm$^3$) | 189 | 0.37 | 0.03 | 0.50 | 0.31 | 8.1 |
| MOE (MPa) | 189 | 4224 | 444.80 | 5776 | 3273 | 10.5 |
| MOR (MPa) | 189 | 49.42 | 5.47 | 64 | 24 | 11.1 |
| CS (MPa) | 189 | 35.52 | 4.29 | 47 | 22 | 12.1 |

*n*: number of samples; SD: standard deviations; Max: maximum value; Min: minimum value; CV%: coefficient of variation; RLMS: released longitudinal maturation strains; FL: fiber length; 2WT: double-wall thickness; MFA: microfibril angle; PG: proportion of G-layer; BD: basic density; MOE: bending modulus of elasticity; MOR: bending modulus of rupture; CS: compressive strength.

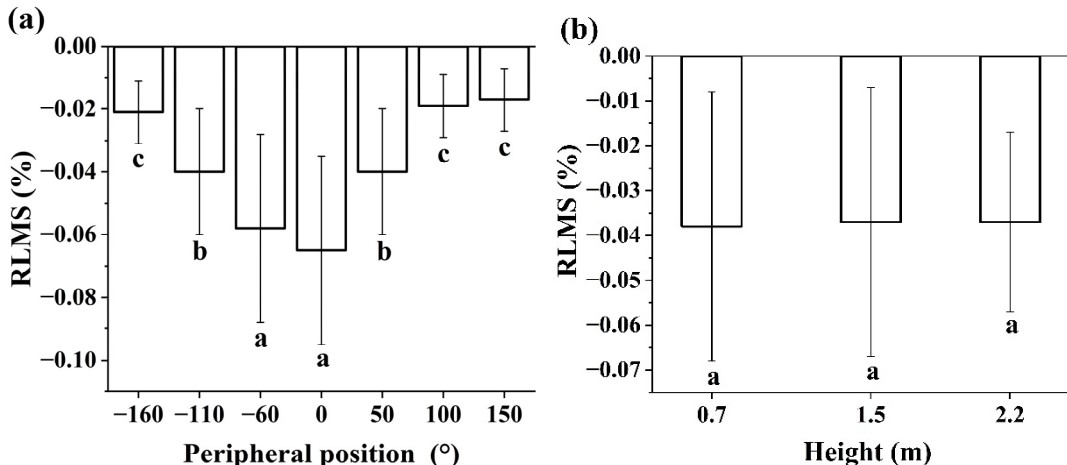

**Figure 4.** Distributions of RLMS with peripheral positions (**a**) and heights (**b**). Different letters indicate significant differences according to Duncan's means comparison tests in $p < 0.05$. RLMS: released longitudinal maturation strains.

### 3.2. Anatomical Properties

The anatomical parameters from the nine sampled trees are listed in Table 2. The CV for the MFA was 45.8%, with the minimum (0.1°) being much lower than the average (14.9°). The mean values for the PG ranged between 0.0% and 68.8%, with the highest CV being 110.9%. There was significant variation in the FL, 2WT, MFA, and PG ($p < 0.001$) with peripheral position around the trunk (Figure 5). The MFA was significantly lower on the upper side of the trunk than on the underside. The upper side of the trunk showed a high PG. The FL and 2WT did not show any specific trends with position.

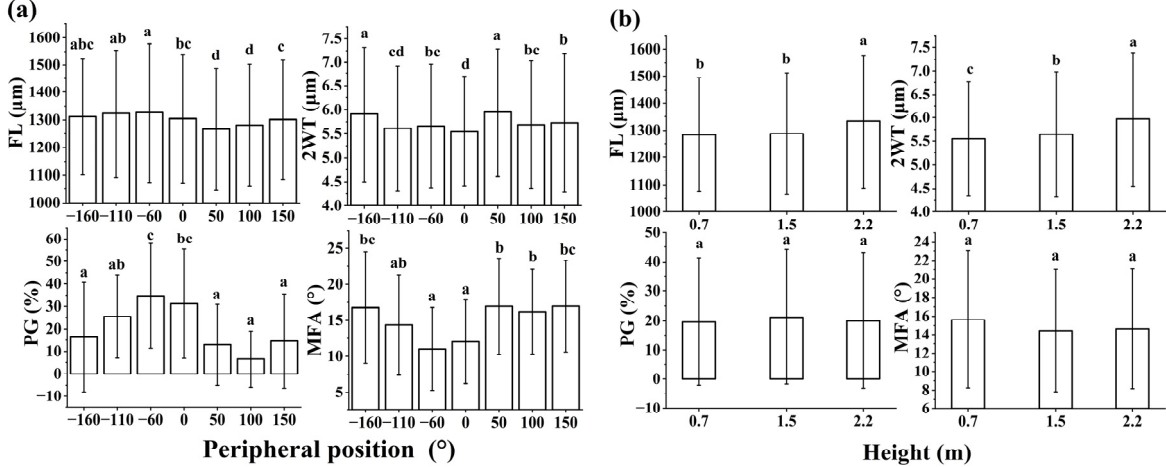

**Figure 5.** Distributions of anatomical measured parameters with peripheral positions (**a**) and heights (**b**). Different letters indicate significant differences according to Duncan's means comparison tests in $p < 0.05$. FL: fiber length; 2WT: double-wall thickness; MFA: microfibril angle; PG: proportion of G-layer.

There was significant variation in the FL and 2WT ($p < 0.01$) with height, except in the MFA and PG ($p > 0.05$). At heights between 0.7 m and 2.2 m, the 2WT increased with height, but other anatomical parameters fluctuated or changed little with height.

### 3.3. Physical and Mechanical Properties

The physical and mechanical properties of wood from the sampled trees are listed in Table 2. There were significant differences in the MOE among positions ($p < 0.001$) but no significant differences in the BD, MOR, and CS ($p > 0.05$). The MOE values were

significantly higher on the upper sides of the stem (Figure 6). The BD, MOR, and CS fluctuated or showed little change with positions. With the increase in height, the BD, MOE, MOR, and CS showed little variation. There were no significant differences in the BD, MOE, MOR, and CS among heights ($p > 0.05$).

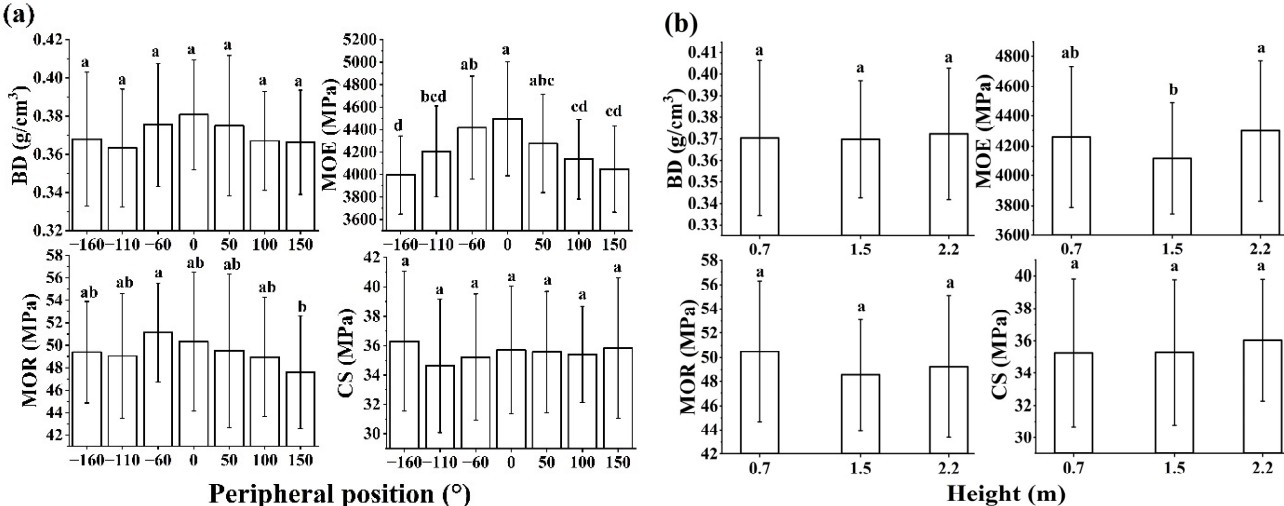

**Figure 6.** Distributions of physical and mechanical properties with peripheral positions (**a**) and heights (**b**). Different letters indicate significant differences according to Duncan's means comparison tests in $p < 0.05$. BD: basic density; MOE: bending modulus of elasticity; MOR: bending modulus of rupture; CS: compressive strength.

### *3.4. Indentation Modulus of Cell Wall*

The indentation modulus of each wall layer in wood fiber cells in the TW (tension wood area), LW (lateral wood area), and OW (opposite wood area) was characterized with AFM using the PF-QNM mode. As can be seen in Figure 7, the fibers of the TW and the LW contained a G-layer, but no G-layer was found in OW cells. There were clear differences in indentation modulus among the three layers (Table 3, Figure 7). In the TW and the LW, the average indentation modulus of the G-layer was higher than that of the S2 and S1 layers. In the OW, from which the G-layer was absent, the average indentation modulus of the S2 layer was larger than that of the S1 layer. The average indentation modulus of the TW was higher (13.9 GPa) than that of the LW (13.3 GPa) and the OW (10.4 GPa). The cell-wall indentation modulus was significantly different between the TW, LW, and OW ($p < 0.05$).

**Table 3.** Means and standard deviations of indentation modulus (in GPa) of cell wall at different areas.

|  | TW | LW | OW |
|---|---|---|---|
| S1 | 9.5 ± 1.41 [a] | 9.4 ± 1.24 [b] | 8.6 ± 1.52 [c] |
| S2 | 15.6 ± 2.51 [a] | 15.1 ± 2.53 [b] | 12.2 ± 2.60 [c] |
| G | 16.6 ± 3.15 [a] | 15.5 ± 2.98 [b] | N.A. |

Different letters indicate significant differences according to Duncan's means comparison tests in $p < 0.05$.

The indentation modulus of the cell wall at each height—0.7 m, 1.5 m, and 2.2 m—is shown in Figure 8, and the indentation modulus values for each wall layer are summarized in Table 4. Since these samples were chosen from position 0°, the G-layers could be observed. There was variation among the S1, S2, and G-layers, and the average indentation modulus of the cell wall at 0.7 m was slightly higher than at 1.5 m and 2.2 m, but the difference was not significant between heights ($p > 0.05$).

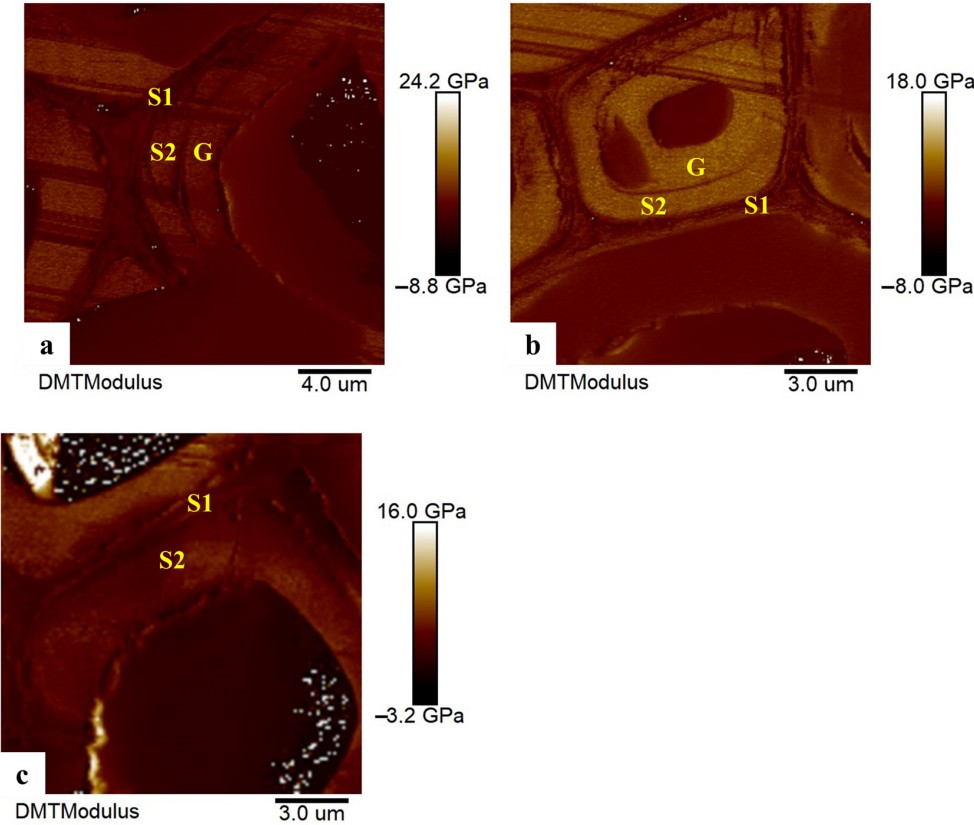

**Figure 7.** Indentation modulus of cell-wall layer in DMT mode. (**a**) TW area, (**b**) LW area, (**c**) OW area.

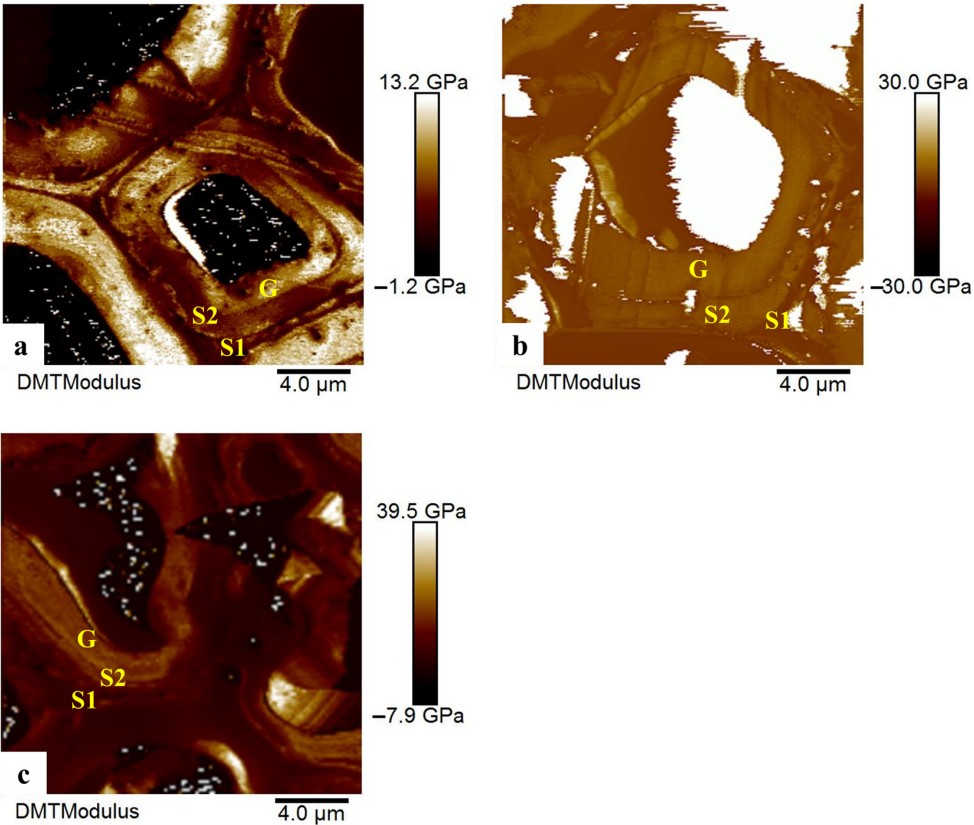

**Figure 8.** Indentation modulus of cell-wall layer in DMT mode. (**a**) 0.7 m, (**b**) 1.5 m, (**c**) 2.2 m.

**Table 4.** Means and standard deviations of indentation modulus (in GPa) of cell wall at different heights.

|  | 0.7 m | 1.5 m | 2.2 m |
|---|---|---|---|
| S1 | 9.4 + 1.98 [a] | 9.3 + 1.59 [a] | 9.2 + 1.43 [a] |
| S2 | 12.1 + 2.35 [a] | 11.3 + 2.44 [a] | 11.6 + 2.53 [a] |
| G | 13.1 + 3.21 [a] | 12.8 + 2.38 [a] | 13.0 + 2.57 [a] |

Different letters indicate significant differences according to Duncan's means comparison tests in $p < 0.05$.

*3.5. Relationships between Released Longitudinal Maturation Strains and Wood Properties*

Pearson correlation coefficients between the RLMS and the wood properties for peripheral positions were calculated and are listed in Table 5. In the TW zone, a positive correlation was found between the RLMS and the PG, BD, MOE, MOR, and IM, and a significant negative correlation was found between the RLMS and the MFA. In the LW zone, a significant positive correlation was found between the RLMS and the FL and BD, and a significant negative correlation was found between the RLMS and the CS. In the OW zone a significant positive correlation was found between the RLMS and the MFA, and a significant negative correlation was found between the RLMS and the PG. There were no significant correlations between the RLMS and other wood properties.

**Table 5.** Results of correlation analysis between RLMS and wood properties among different peripheral positions.

|  | TW (0°, 50°, −60°) | | | LW (100°, −110°) | | | OW (150°, −160°) | | |
|---|---|---|---|---|---|---|---|---|---|
|  | *p* | Sig. | df | *p* | Sig. | df | *p* | Sig. | df |
| FL | −0.071 | ns | 81 | 0.270 | * | 54 | 0.185 | ns | 54 |
| 2WT | 0.14 | ns | 81 | 0.146 | ns | 54 | 0.176 | ns | 54 |
| PG | 0.386 | ** | 81 | 0.234 | ns | 54 | −0.290 | * | 54 |
| MFA | −0.482 | ** | 81 | −0.078 | ns | 54 | 0.323 | * | 54 |
| BD | 0.267 | * | 81 | 0.340 | * | 54 | −0.169 | ns | 54 |
| MOE | 0.322 | ** | 81 | 0.205 | ns | 54 | 0.144 | ns | 54 |
| MOR | 0.303 | ** | 81 | 0.144 | ns | 54 | 0.159 | ns | 54 |
| CS | 0.027 | ns | 81 | −0.269 | * | 54 | −0.137 | ns | 54 |
| IM | 0.955 | ** | 9 | 0.523 | ns | 9 | 0.643 | ns | 9 |

RLMS: released longitudinal maturation strains; FL: fiber length; 2WT: double-wall thickness; PG: proportion of G-layer; MFA: microfibril angle; BD: basic density; MOE: bending modulus of elasticity; MOR: bending modulus of rupture; CS: compressive strength; IM: indentation modulus of cell wall; ns: non-significant; *: significant at 0.05 level; **: significant at 0.01 level.

Pearson correlation coefficients between the RLMS and the wood properties at different heights were calculated and are listed in Table 6. At the height of 0.7 m, a significant positive correlation was found between the RLMS and the 2WT, PG, BD, MOE, and MOR, and a significant negative correlation was found between the RLMS and the MFA. At the height of 1.5 m, a significant positive correlation was found between the RLMS and the FL, PG, BD, MOE, and MOR. At the height of 2.2 m, a significant positive correlation was found between the RLMS and the MOE, and a significant negative correlation was found between the RLMS and the IM. There were no significant correlations between the RLMS and other wood properties.

**Table 6.** Results of correlation analysis between RLMS and wood properties at different heights.

| | 0.7 m | | | 1.5 m | | | 2.2 m | | |
|---|---|---|---|---|---|---|---|---|---|
| | *p* | Sig. | df | *p* | Sig. | df | *p* | Sig. | df |
| FL | −0.081 | ns | 63 | 0.265 | * | 63 | −0.059 | ns | 63 |
| 2WT | 0.345 | ** | 63 | 0.065 | ns | 63 | 0.023 | ns | 63 |
| PG | 0.312 | * | 63 | 0.393 | ** | 63 | 0.24 | ns | 63 |
| MFA | −0.393 | ** | 63 | −0.179 | ns | 63 | −0.075 | ns | 63 |
| BD | 0.258 | * | 63 | 0.338 | ** | 63 | 0.159 | ns | 63 |
| MOE | 0.429 | ** | 63 | 0.497 | ** | 63 | 0.352 | ** | 63 |
| MOR | 0.444 | ** | 63 | 0.252 | * | 63 | 0.066 | ns | 63 |
| CS | −0.109 | ns | 63 | −0.105 | ns | 63 | −0.04 | ns | 63 |
| IM | 0.647 | ns | 9 | 0.005 | ns | 9 | −0.773 | * | 9 |

RLMS: released longitudinal maturation strains; FL: fiber length; 2WT: double wall thickness; PG: proportion of G-layer; MFA: microfibril angle; BD: basic density; MOE: bending modulus of elasticity; MOR: bending modulus of rupture; CS: compressive strength; IM: indentation modulus of cell wall; ns: non-significant; *: significant at 0.05 level; **: significant at 0.01 level.

## 4. Discussion

Wilson and Gartner [42] reported that growth stress increased with lean on the top side, and Yoshida et al. [43] found that growth stress increased from 0° (vertical) to 20°. As shown in Figure 9a, the effect of inclination angle on maturation stress in artificially inclined poplar trees is limited ($p > 0.05$), which is in contrast to findings from previous studies. Analysis of the relationship between the mean RLMS on a tree and its eccentricity reveals significant relationships ($p < 0.05$), as shown in Figure 9b. This means that eccentric growth increases the magnitude and heterogeneity of tensile RLMS. Eccentric growth, wood stiffness, and growth stress all increase the efficiency of controlling the orientation of tree axes [8,43]. Our results support the hypothesis that gravitropism (ability to restore upright growth) in poplar trees is synergistically affected by increased tensile growth stress and promotion of secondary growth on the upper side of the inclined stem [44]. The mean values for the RLMS (−0.16%–0.00%) here are similar to those found in other studies on *Populus* × *euramericana* [15] and *Eucalyptus grandis-urophylla* [45].

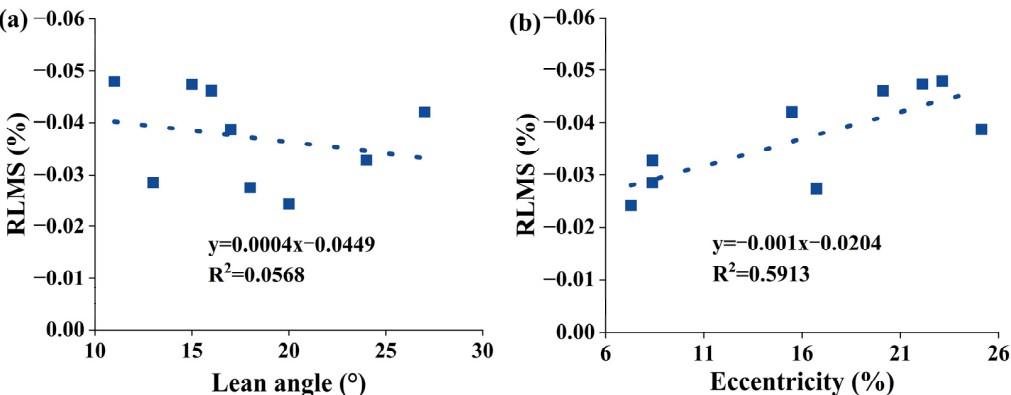

**Figure 9.** (**a**) Relationship of RLMS to lean angle in inclined trees, regression coefficients were not significant ($p = 0.53$). Mean lean angle and SD = 17.9° ± 5.11. (**b**) Relationship of RLMS to eccentricity in leaning trees, regression coefficients were significant ($p = 0.01$). Mean eccentricity and SD = 16.3% ± 6.90.

The RLMS were significantly different between peripheral positions, which is interpreted as an adaptation enabling them to re-orient the stem to vertical growth [44]. The effect of sampling height was not significant and was consistent with findings by Li et al. [30] for *Populus* × *euramericana* cv. '74/76' inclined trees. Huang et al. [46] also found no obvious relationship between height in tree and measured surface strains in the axial direction in *Zelkova serrata* erect trees. Other studies suggest that growth stresses do

vary with sampling height, e.g., Omonte and Valenzuela [47] found that growth stress indicators at different heights were highly variable in the case of *Eucalyptus nitens* erect trees. Naghizadeh and Wessels [45] reported that growth strains decreased from 1.4 m to 11.4 m in *Eucalyptus grandis-urophylla* erect trees.

Results from this study found no significant difference in fiber length between tension wood and normal wood, in accordance with findings by Scurfield and Wardrop for *Acacia acuminata* Benth. and *Casuarina cristata* Miq. [48]. Fang et al. [17] reported increased cell-wall thickness in mild and severe tension wood. However, in this study, the 2WTs in the upper-side tension wood were similar to those in cells from the other positions. The MFAs range from 15°–30° in the S2 layer, and 0°–2.5° in the G-layer [49]. Our results accordingly showed low MFAs on the upper side and larger MFAs on the lower side of the poplar trunks. The smaller the microfibril angle, the larger the longitudinal tensile stress [7,50,51]. The PG has been shown to be greater on the upper side, which can trigger higher tensile stresses for the vertical recovery of stems [2,52].

In accordance with previous works, our findings showed that the BD remains unchanged between tension and normal wood [53]. Higher MOE, MOR, and CS on the upper side has been observed in ten tropical rainforest trees including *Cecropia sciadophylla* [51]. In this study only the MOE followed the same trend, but the MOR and the CS showed minimal change. The higher wood MOE and indentation modulus of the cell wall in TW are believed to be related to the characteristics of the G-layer, which include higher cellulose content and crystallinity, and lower microfibril angles [24].

Li et al. [54] found that with an axial height from 0 m to 11.3 m in *Catalpa bungei* trees, fiber wall thickness decreased. Wu et al. [23] found that in triploid hybrids of *Populus tomentosa* fiber length decreased with height of between 2 m and 6 m. De Boever et al. [55] observed an increase in wood MOE, MOR, and density with tree height in hybrid poplar trees from 1.2 m, 6.5 m, and 11.5 m, which they attributed to changes in the ratio of heartwood to tension wood. Kijidani and Kitahara [56] found the MOE of *Cryptomeria japonica* wood to be greater and the MFA lower at 5 m height compared to at 1.5 m, while the wood BD and latewood tracheid length did not change with height. Lima Jr. et al. [57] found that the MOE of *Eucalyptus grandis* wood increases with height of 3 m, 6 m, and 9 m from the base. Himes et al. [58] found that the density, MOR, and MOE of hybrid poplar lumber samples generally increased with height from 0 m to 10 m. In this study, lower heights (0.7 m, 1.5 m, and 2.2 m from the ground) were selected for the RLMS and wood properties than in other studies due to restricted electric-wire length and the operational safety requirements of measuring RLMS in living trees. Here, only the 2WT was found to increase with height, while the FL and MOE fluctuated with no particular relationship with height, and the RLMS, MFA, PG, BD, MOR, CS, and indentation modulus of the cell wall did not change with sampling height in tree. Increased 2WT with height may be attributed to the thinning treatment over the 12-year growth period [59].

Many researchers have reported the correlations between growth stress/growth strain and wood properties [30,60], but the effects and correlations among different peripheral positions and heights are seldom reported or discussed. Correlations in the TW zone were stronger than in other zones. It should also be noted that the correlations at the different heights were quite different: there were significant correlations between the RLMS and the wood properties when measured at 0.7 m and 1.5 m, while at 2.2 m correlations weakened considerably. Therefore, RLMS could be considered good indicators of wood properties in the TW area of the trunk below 1.5 m height.

## 5. Conclusions

The influence of peripheral positions and heights on the released longitudinal maturation strains (RLMS), anatomical features, physical and mechanical properties, and nano-mechanical properties of the cell wall were assessed in this work. Correlations between the RLMS and the wood properties were also analyzed. The results showed that the upper sides of the inclined trunks had higher RLMS, proportion of G-layer, bending

modulus of elasticity, and indentation modulus of the cell wall but a lower microfibril angle than the other sides. These differences may be related to the existence of the G-layer in the tension wood area. The sampling height in the trunk had little effect on measured wood and cell characteristics, except for double-wall thickness, which tended to be lower closer to the base of trees (when measured at 0.7 m, 1.5 m, and 2.2 m from the ground). RLMS showed strong correlations with wood characteristics in the tension wood zone and with height of up to around 1.5 m.

In general, the artificial inclination in the trees had mixed effects on wood properties and quality. The variation observed in this study could be used to improve knowledge of a phenomenon that influences the production of quality wood and makes it less suitable for final destinations with higher added value. Additional studies are required to evaluate the variability in longitudinal maturation strains and wood characteristics higher up in the trunk for appropriate selection strategies.

**Author Contributions:** Conceptualization, Y.L. and X.W.; methodology, Y.L.; software, J.Z.; validation, C.D. and S.L.; formal analysis, Y.L.; investigation, X.W.; resources, S.L.; data curation, X.W.; writing—original draft preparation, Y.L.; writing—review and editing, K.S.; visualization, J.Z.; supervision, K.S. and C.D.; project administration, X.W.; funding acquisition, Y.L. All authors have read and agreed to the published version of the manuscript.

**Funding:** This research was funded by the China Scholarship Council (202008775007) and the Natural Sciences and Engineering Research Council of Canada (RGPIN-2020-06097).

**Data Availability Statement:** Data is unavailable due to privacy.

**Acknowledgments:** We acknowledge N.P., S.F, M.Z. and X.W. for the technical assistance they provided.

**Conflicts of Interest:** The authors declare no conflict of interest.

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
