# Peer review of "Maturation Stress and Wood Properties of Poplar (Populus × euramericana cv. ‘Zhonglin46’) Tension Wood"

_forests, doi:10.3390/f14071505_

Round 1

Reviewer 1 Report

In the chapter “Introduction” I suggest to better explain that maturation growth stresses are also present in normal wood tissues while paper deals in particular to their effects in the portion of the stem interested by tension wood. Moreover, since tension wood generally occurs locally and only concerns some growth rings, it should be mentioned how the Authors recognized/detected (visually?) this type of reaction wood in order to perform the experimental measurements to its level.

Specific comments for the Authors are include as highlighted text in the attached pdf.

Reviewer 2 Report

Solid study. Could be useful for other researchers.

This work appears to be a reliable report on the conducted physicochemical analyses of poplar wood. The proposed analytical techniques have been well-known and used in forestry for decades. The scope of the obtained information is also not novel. The studied material is widely described in the literature, but the article lacks reference to similar results from other researchers, e.g. on other species. Nevertheless,  the work contains much information about one research material in one place. 

Reviewer 3 Report

Comments and Suggestions for Authors

Dear Authors and Editors,

This paper focuses on the influence of peripheral positions and heights on released longitudinal maturation strains, anatomical features, physical and mechanical properties, and nanomechanical properties of the cell wall in poplar tension wood.

The results of this research are original, significant, and well-defined, and all conclusions are justified and supported by the results.

The authors provided an introduction of pre-research, but it could be expanded with more information on tension wood and its formation in trees and more about poplar species.

For future research, it would be advisable to measure properties at greater differences in height and between the trees of closer diameter range.

The tests were provided according to national standards, but the methodology needs some more explanation. The results are original and statistical analysis is provided in significant volume.

English language is understandable and needs some minor corrections.

The paper needs minor corrections and some observations in the text below.

Line 28. You should expand the introduction with more information on tension wood and its formation in trees and more about poplar species.

Here is some literature for consideration:

Panshin, A.J., de Zeeuw, C. (1964): Textbook of Wood Technology: Structure, Identification, Properties, and Uses of the Commercial Woods of the United States and CanadaVol. 4, McGraw-Hill, United States of America.

Tsoumis, G. (1991) Science and technology of wood: structure, properties, utilization. Chapman & Hall, New York, 66-83

Zobel, B.J., van Buijetenen, J.P. (1989) Wood variation. Its causes and control. Springer-Verlag, Berlin, Heildeberg.

DeBell, D. S., Singleton, R., Harrington. C.A., Gratner, B.L. (2002) Wood density and fiber length in young Populus stems: relation to clone, age growth rate and pruning. Wood and Fiber Science, 34, 529-539.

Ištok I, Potočić N, Šefc B, Sedlar T. The Effects of Nitrogen Fertilisation on the Anatomical Properties of the Populus alba L. Clone ‘Villafranca’ Juvenile Wood. Biology. 2022; 11(9):1348. https://doi.org/10.3390/biology11091348

Line 86. Table 1. Is the diameter of second tree correct (51,2 cm)? If it is, it would be great to calculate difference between trees or between diameter classes, in some other manuscript.

Line 120. Use a section break at the end of the first sentence to split these methods.

Line 124. Table 3 a

Line 127. What were the dimensions of the areas?

Line 127. You should use a citation at the end of the sentence about double stained method.

Line 131. Which instrument did you use for the measurement of X-ray diffraction?

Line 134. This is the first time you have used the abbreviation BD in text. What is BD?

Line 199. How do you explain this?

Line 206. How do you explain that there is no difference between BD in tension wood and opposite wood?

When you started the test, you divided the cross-section of the stem into three groups (TW, LW OW). Why didn’t you base your calculation on the differences between those three groups? Yet you calculated the mean value of 9 samples from 9 different trees at the same position according to the inclination of the stem. 

English language is understandable and needs some minor corrections.
